# Mesocosm experiments in ocean alkalinity enhancement research

Ulf Riebesell[1], Daniela Basso[2], Sonja Geilert[3], Andrew W. Dale[1], Matthias Kreuzburg[4]

[1]GEOMAR Helmholtz Centre for Ocean Research Kiel, Kiel, Germany
[2]Department of Earth and Environmental Sciences, University of Milano-Bicocca, Milan, Italy,
[3]Geosciences, Utrecht University, Utrecht, The Netherlands
[4]Department of Biology, University of Antwerp, Antwerp, Belgium
*Correspondence to*: Ulf Riebesell (uriebesell@geomar.de)

**Abstract.** An essential prerequisite for the implementation of ocean alkalinity enhancement (OAE) applications is their environmental safety. Only if it can be ensured that ecosystem health and ecosystem services are not at risk will the implementation of OAE move forward. Public opinion on OAEs will depend first and foremost on reliable evidence that no harm will be done to marine ecosystems and licensing authorities will demand measurable criteria against which environmental sustainability can be determined. In this context mesocosm experiments represent a highly valuable tool in determining the safe operating space of OAE applications. By combining biological complexity with controllability and replication they provide an ideal OAE test bed and a critical stepping stone towards field applications. Mesocosm approaches can also be helpful in testing the efficacy, efficiency and permanence of OAE applications. This chapter outlines strengths and weaknesses of mesocosm approaches, illustrates mesocosm facilities and suitable experimental designs presently employed in OAE research, describes critical steps in mesocosm operation, and discusses possible approaches for alkalinity manipulation and monitoring. Building on a general treatise on each of these aspects, the chapter describes pelagic and benthic mesocosm approaches separately, given their inherent differences. The chapter concludes with recommendations for best practices in OAE-related mesocosm research.

## Preface

The authors would like to emphasize that this chapter does not intend to cover all aspects of mesocosm experimentation in its full breadth, but rather tries to address aspects specific to research on ocean alkalinity enhancement (OAE) or aspects we consider important to reiterate here. For a more comprehensive presentation of recommendations and guidelines on mesocosm experiments the reader is referred to Chapter 6 of the *Guide for Best Practices on Ocean Acidification Research and Data Reporting* (Riebesell et al. 2010) and references therein as well as Stewart et al. (2013).

Although the general approach to mesocosm experiments is straightforward and basically involves enclosing a body of water with or without sediment in order to monitor responses of the enclosed communities and related

processes to the manipulated perturbation over an extended period of time, the specifics of conducting such
experiments can vary considerably. These include factors such as the materials, design and location of the
enclosures, e.g. fixed structures on land or flexible wall enclosures in situ, as well as the procedures for mesocosm
filling, operation, mixing and sampling. While the dimensions of the experimental enclosures can range from less
than 1 $m^3$ to >1000 $m^3$ depending on the requirements of the experiment, we here adopt the classification set out
by the SCOR Working Group 85 in 1991: Microcosms (less than 1 $m^3$), mesocosms (between 1 and 1000 $m^3$) and
macrocosms (more than 1000 $m^3$). We note that benthic experimental enclosures can have different size
categories.

## 1 Placing mesocosms in the context of OAE research

Mesocosm experiments provide an essential bridge between the tightly controlled but poorly realistic laboratory
culture experiments and the complexity of natural systems. This is particularly important for possible OAE
implementations, in order to achieve a sound understanding of the entire process of the proposed OAE strategies,
from the dissolution kinetics and effectiveness of the alkalinisation technique, to the potential environmental
impacts, risks and co-benefits. This knowledge is crucial prior to any form of OAE application to safeguard the
protection of marine ecosystems functioning, biodiversity and related ecosystem services. Moreover, should OAE
prove to be a viable approach for marine carbon dioxide removal (mCDR), it will also be crucial to achieve social
acceptance for potential OAE implementations. Also in this context mesocosm experiments can serve as a useful
tool for proof of concept, the results of which can play an important role in the public discourse about the risks
and benefits of mCDR implementation.
Functional redundancy and species richness in ecosystems allow for some degree of resistance to withstand
disturbances and resilience to recover once a disturbance has ended or dissipated. To determine the actual
ecological impacts of OAE it is essential, therefore, to test suggested applications at the community/ecosystem
level. Doing this in field trials, however, poses serious difficulties, given the hydrographic complexity of most
marine systems, with lateral advection (currents, tides), vertical flow (convection, up- and downwelling) and
wave-driven mixing. Determining dose-response relationships for environmental impacts is extremely
challenging under such conditions. Mesocosm experiments, on the other hand, enable the combination of
biological complexity needed for testing resistance and resilience of communities/ecosystems in their natural
setting and seasonal succession (in a single experiment where succession occurs on short time scales, e.g. a
phytoplankton bloom, or multiple experiments in different seasons using the exact same experimental set-up) with
a reasonable degree of control and replication and hence the statistical power to reach reliable conclusions. At the
same time, they allow testing the chemical kinetics of mineral dissolution and secondary carbonate precipitation,
thereby providing vital information on the efficacy of the suggested OAE applications in a natural setting under a
range of environmental conditions (salinity, temperature, carbonate chemistry, inorganic nutrient concentrations,
dissolved and particulate organic carbon concentrations etc). Testing them in mesocosm enclosures has the
additional benefit of minimizing public concern and regulatory requirements when compared to field trials.
Environmental impacts of OAE will be scale- and context-dependent in terms of the physical (e.g. timescales of
mixing and $CO_2$ equilibration, point source vs. diluted release), chemical (e.g. amount/type of alkaline substance,
impurities), and biological characteristics (e.g. seasonal succession and related ecosystem vulnerability).
Biological impacts are determined by exposure time and dose, ranging from acute shock responses on transient
and local scales at point sources to chronic effects associated with possible transitions of ecosystem structure and
performance at the regional and long-term scale. Key research questions which can be addressed adequately
through mesocosm experiments are:
- What is the safe operating space for OAE applications with respect to possible impacts on marine
ecosystems functioning, biodiversity, and ecosystem services?
- How could OAE be implemented to reduce the risk of inadvertent negative environmental effects, and
maximize co-benefits?
- Which biological indicators can serve as early warning signals or proxies for OAE environmental
impacts?
- How do different OAE approaches perform in terms of efficiency (e.g. mineral dissolution, $CO_2$ uptake)
and permanency (e.g. secondary precipitation)?
- Which application sites are most appropriate for which OAE approach?

## 2 Strengths and weaknesses of mesocosm experimentation

Mesocosm experiments offer a salient advantage over laboratory-based investigations, as they allow a realistic
replication of natural communities. Multiple trophic levels can be confined under natural environmental
conditions over a long period of time in a self-sustaining manner. Thereby, the same community can be sampled
repeatedly over time. Furthermore, these experiments permit straightforward validation in the context of field
research. Mesocosms, in essence, are closer to representing natural ecosystems characterized by carefully defined
dimensions and monitored conditions and processes. To ensure realistic ecological boundary conditions,
mesocosm experiments should be exposed to meteorological conditions resembling those of the target
environment. Notably, the logistical flexibility of mesocosms affords researchers the opportunity to conduct
investigations beyond the geographical confines of the environment under investigation. Consequently,
mesocosms provide an invaluable avenue for the controlled study of specific environments and the impact of
controlled manipulations therein. Given the diverse range of natural processes encountered in mesocosm
experiments, external influences may be challenging to control, necessitating a robust monitoring strategy to
achieve statistical power by either treatment replication or treatment gradients. Moreover, mesocosm experiments
provide extensive multidisciplinary datasets that allow for a high degree of scientific integration and
interdisciplinary collaboration. These datasets are valuable for parameterisation and assessment of marine
ecosystems and biogeochemical models.
While mesocosm experiments can be considered the preferred tool for the assessment of environmental impacts
of OAE applications, they have several weaknesses that need to be considered when interpreting the data and
extrapolating the results to the real world. These weaknesses include unnatural mixing and turbulence (in pelagic
mesocosm), unnatural flow of bottom water across the sediment (in benthic mesocosms), wall effects and the
growth of periphyton and other organisms on the mesocosm walls, spatial heterogeneity in the enclosed sediments
and the related difficulties in obtaining representative samples. The larger and more expensive the enclosures
become, the more difficult it becomes to have a sufficient number of replicates in a replicated design or treatments
in a gradient design. The fact that even the largest mesocosms enclose truncated communities, i.e. exclude higher
trophic levels and highly migratory organisms make it difficult to adequately represent the responses of organisms
with longer life cycles and the associated impacts on the food web. Another drawback of mesocosm experiments
is their limited duration, due to the gradual diversion from their natural counterparts, e.g. due to community shifts,
nutrient depletion, and the consequent progressive loss of biological realism. The increasing variability between
mesocosms in this process makes it increasingly difficult to identify treatment effects with statistical significance.
**3 Experimental design**
The primary purpose of a mesocosm experiment is to obtain "near-natural" conditions, that is to say, keeping the
abiotic and biotic factors as close to the environment as possible in order to maximize the realism of the tested
conditions. In general, time scale is related to mesocosm volume: the shorter the time needed for a controlled
experiment, the smaller the enclosure size. Careful consideration should be given to the experimental design to
adequately address the specific research questions, account for ecosystem- and site-specific characteristics as well
as seasonal variability. The choice of the experimental configuration includes the three key dimensions of time,
space and biological complexity, along with the required level of replication. Preference should be given to mimic
the natural seasonal succession rather than provoking out-of-season events, e.g. triggering phytoplankton blooms
through nutrient addition.
Considering the often limited number of experimental units, a critical consideration concerns the level of
replication (Kreyling et al. 2018). The choice is between two basic approaches: (1) replicated (n≥3) treatments,
with limited treatment levels (e.g. Riebesell et al., 2006); (2) a gradient approach with a larger number of non-
replicated treatment levels (e.g. Taucher et al., 2017). The statistical power of the two options, using ANOVA
statistics for the replicated design and regression statistics for the gradient design, is similar for the small number
of experimental units typically available in mesocosm studies (Havenhand et al., 2010). If large within-treatment
variation is expected, e.g. due to strong environmental variability or spatial heterogeneity, the replicated approach
is recommended. In fact, strong within-treatment variability can easily mask subtle treatment effects. An important
advantage of the gradient approach, on the other hand, is that it enables the identification of non-linearities,
thresholds and tipping points in biological responses to OAE applications, relevant information for model
parameterizations in terms of community functional responses. Knowledge about thresholds and possible tipping
points is crucial also in the context of regulatory considerations for OAE implementation.
Pelagic mesocosms
When aiming to investigate OAE applications in the free water column, pelagic mesocosms are the research tool
of choice. Among the various proposed strategies, ocean liming in the wake of ships would consist of sparging
high-alkalinity fluids or mineral particles within the surface layer in offshore settings. In this scenario, any
chemical perturbation is expected to affect in the first instance the pelagic domain and the planktic component of
the marine ecosystem. Also OAE applications at fixed locations with a discharge of alkalinity-enriched water into
coastal waters, e.g. desalination plants or sewage treatment plants, are best simulated in pelagic mesocosms. A
suitable simulation of OAE approaches in which the alkalising mineral is released in particulate form should
ideally have the dissolution rate of the particles known in advance. If the rate is fast enough to ensure complete
dissolution in the water column, pelagic mesocosms are well suited. In cases where the dissolution rate is slow
compared to the particle sinking rate and particles sink to the seabed before dissolving, the experimental design
may require a benthic component.
A missing component in all closed-system mesocosm experiments is the dilution through mixing with non-
perturbed waters. Switching to an open system, where the enclosed water is partially replaced by non-alkalised
water, places much greater demands on monitoring and complicates the interpretation of the observed responses,
to the extent that it may be impossible to establish a reliable dose-response relationship. This experimental artifact
is exacerbated when repeated additions of alkalinity are applied. Incorporating naturally occurring dilution in the
experimental design can be done by applying the OAE treatment to only part of the enclosed water column and
allowing for gradual mixing with the untreated water. The time until mixing can be controlled by stratifying the
water column through a salinity gradient (adding fresh water into the upper layer or brine into the bottom layer,
whereby the salinity change should be at a low enough level not to cause a biological response, e.g. a few tens of
a salinity unit) or via a temperature stratification. Break-off of the stratification can be gradual or abrupt through
active mixing. Parallel sampling of the OAE treated and untreated water bodies can provide insights about the
compensating effect of dilution.
There is a wide range of enclosure volumes and structures used in pelagic mesocosm experimentation. Among
the various available solutions, the most obvious difference is the placement of the mesocosm: 1) stable,
permanent structures on land, or 2) floating bags in the water. All materials that come into contact with the
enclosed water/sediment must be chemically inert, i.e. they must not leach or actively absorb any substances.
Some technical details of the mesocosm design can markedly affect some abiotic factors, such as thermal
characteristics, light conditions or mixing intensity of the enclosed water column. Most pelagic mesocosm
enclosures are made of transparent material supported by a mini-mal rigid framework, with the intent to keep light
conditions as in nature. Most materials, however, change the spectrum of the transmitted light, for example are
not transparent for UV-light. As enclosure depth is often lower than the mixed layer depth of the natural
environment, natural light conditions are not well represented in mesocosms, with light intensities averaged over
the mesocosm depth often higher than those averaged over the mixed layer depth.
Benthic mesocosms
Benthic mesocosm experiments offer the unique chance to study OAE-mineral addition to the seafloor in a
controlled set-up. In comparison to experiments in laboratory settings, often small in scale with respect to mineral
weathering, benthic mesocosms are more likely to mimic natural seafloor conditions and allow the coupling of
biogeochemical processes at larger spatial and temporal scales. Key research questions on seabed alkalinisation
to be addressed in benthic mesocosm experiments include: 1) What are alkaline mineral dissolution rates under
mesocosm conditions? 2) Do secondary minerals form that may compromise the net $CO_2$ sequestration efficiency
of this method? 3) How are microbial communities and macrofauna affected by mineral dissolution? 4) Is there a
release and accumulation of heavy metals related to addition of silicate-based minerals and how does their toxicity
affect the community/ecosystem?

Continuous water flow system: In this set-up, a continuous flow of ambient seawater, preferably bottom water,
over the sediment (Fig. 2), likely best resembles natural seafloor conditions. It is recommended to remove larger
debris that could obstruct the water supply using a sediment trap (Fig. 2), whilst allowing small particulate matter
to enter the mesocosms. The supply of particulate matter is essential to sustain natural microbial metabolism in
the sediments and to provide food for filter-feeding macrofauna that colonize the sediment surface within a short
period of weeks to months (Fig. 2). A relatively high flow rate is required (between 5000 to 10000 L $d^{-1}$) to keep
the seawater well oxygenated and guarantee the survival of fauna and for maintaining the natural microbial
communities as closely as possible to in situ conditions. With this set-up, the bottom water should be monitored
to trace seasonal changes in physical and chemical properties of the incoming seawater.
Water circulation approach: The benthic mesocosm set-up with a seawater circulation approach consists of two
tanks stacked on top of each other, with the upper tank housing the benthic ecosystem with sediments and
organisms and the lower tank is functioning as a seawater reservoir from which water is pumped into the upper
tank (Fig. 3). Thus, a constant flow of water is generated through the water in- and outflow and the height of the
water column in the upper tank can be controlled by the vertical positioning of the outflow. The tanks for the
benthic mesocosms have a volume of approximately 1 $m^2$ and are situated outdoors and exposed to natural
temperature fluctuations.
Based on the water circulation approach, the closed system allows for the detection and accumulation of
weathering products and to focus on a specific process or reaction, such as the dissolution kinetics of silicate
minerals in the case of the University of Antwerp study (Fig. 3). After a defined timespan (flux session) the total
amount of water is replaced and accumulation of weathering products starts again from initial values. In terms of
this experiment design, ≥3 replicates of benthic mesocosms are crucial to ensure that results are statistically
significant and can be generalized to the broader ecosystem being studied (e.g. Wadden Sea).
The total experiment duration as well as the sampling strategy is defined by the research questions and longer
experiments may be necessary to capture seasonal or long-term trends in the system. The use of natural sediment
and the inclusion of a dominant bioturbating organism (e.g. *Arenicola marina*) in benthic mesocosm experiments
is a crucial step toward making the experimental setup more representative of real-world conditions. However,
it's important to emphasize that the choice of sediment type and benthic organisms should be aligned with the
specific research objectives and questions being addressed.
In OAE studies involving benthic mesocosms, various types of sediments can be considered, ranging from fine-
grained sediments to rocky substrates. The selection of sediment type should be guided by factors such as the
local environmental conditions, the availability of sediment types that reflect the targeted ecosystem, and the
specific geochemical interactions being investigated. For studies related to carbonate dissolution and alkalinity
enhancement as given above, fine-grained or sandy sediments are most suitable, given their potential to facilitate
mineral dissolution and subsequent alkalinity release.
Similarly, the choice of benthic organisms should be tailored to the research objectives. While many benthic
organisms can be tested in mesocosms, it's important to consider the life history, behavior, and ecological role of
the selected species (Bach et al. 2019; Flipkens et al. 2023). For instance, if the experiment spans a year and aims
to study the recruitment and life cycle of benthic organisms that have a pelagic phase, careful planning is required.
Monitoring larval settlement, growth, and interactions with the sediment during their benthic phase becomes
integral to such investigations.
As an illustrative example, consider an OAE study targeting the enhancement of carbonate precipitation through
the addition of alkalinity. In a coastal setting, sandy sediments rich in carbonate minerals might be chosen, given
their potential for mineral dissolution and subsequent bicarbonate formation. Benthic organisms like filter-feeding
mollusks and burrowing polychaetes could be tested to assess their responses to altered alkalinity levels.
Finally, the water circulation approach should be carefully designed to ensure consistency in water flow rates and
initial seawater chemistry. Sedimentation in the water reservoir tank has to be prevented to avoid secondary
sediment surfaces and a continuous monitoring system (salinity, temperature) is recommended to estimate
evaporation rates. In addition, regular sampling of environmental conditions (humidity, $pCO_2$) as well as carbonate
system parameters and nutrients, can ensure that the experiment proceeds as planned and that the results are
reliable.

## 4 Mesocosm operation: filling, sampling, wall cleaning

Filling of the mesocosms is a delicate process that, if not done with care, can jeopardize the entire experiment. A
key aspect is to ensure identical starting conditions, both for the abiotic and biotic conditions in all mesocosms.
Between mesocosm differences in baseline conditions can cause divergence of the enclosed communities and
severely hamper the detection of treatment effects. As the filling often represents a major perturbation itself, some
time of equilibration may be needed before applying the treatment manipulation and starting the actual
experiment. The time for equilibration may differ for pelagic and benthic habitats as well between different
ecosystems and seasons. Adequate monitoring during this pre-manipulation phase can determine when a new
steady state is reached and confirm whether all mesocosms have similar starting conditions. Key parameters for
which equal starting conditions among mesocosms need to be ensured include temperature, salinity, inorganic
nutrient concentrations, the carbonate chemistry (pH, pCO2, DIC TA), dissolved and particulate organic matter
concentrations, community composition and diversity, and standing stocks of the dominant taxonomic groups
across trophic levels.
Another critical aspect of mesocosm operation is taking representative samples. The enclosed water bodies and
sediments typically show spatial heterogeneity (vertical gradients in the water column and sediments, patchiness
in the distribution of larger organisms). The spatial variability of the target variables of the enclosed system should
be determined prior to deciding on the best sampling strategy. Sampling bias related to vertical gradients, e.g.
water column nutrient concentration and phytoplankton biomass, can be overcome by taking depth-integrated
water samples (Fig. 4). Some species may even perform diurnal vertical migration, which also should be accounted
for in the sampling strategy.
Mesocosm enclosures are always associated with additional surfaces, the mesocosm walls, that are not present in
the natural environment. The smaller the mesocosms, the larger the additional surface area relative to the enclosed
volume. Free surfaces are generally subject to rapid biofilm formation, followed by colonization of larger
organisms. The associated microbial community can significantly influence water column processes, which is of
particular concern in pelagic mesocosms. To minimize such wall effects, cleaning of the mesocosm walls can be
useful. Specific to OAE mesocosm experimentation is that under conditions where the water column is highly
oversaturated with respect to calcium carbonate, mesocosm walls can provide free surfaces for secondary
precipitation of carbonates. Under these circumstances, wall cleaning can scrape off these carbonates, creating
additional precipitation nuclei in the water column. If wall cleaning is continued under these circumstances,
possible effects caused by this, e.g. enhancement of secondary precipitation in the water column and increased
ballasting of particulate matter, should be seen as artifacts and interpreted as such. If wall cleaning is discontinued
and the biofilm on the walls grows to a significant biomass compared to the suspended biomass, this may limit
the duration of the experiment. The decision for or against wall cleaning must be made on a case-by-case basis
and depends, among other things, on the severity of wall growth, the duration of the experiment and the specific
research questions to be investigated.
Pelagic mesocosms
Different techniques have been employed for filling pelagic mesocosms, including (1) direct pumping from the
sea in cases where mesocosms are placed *in situ* or close to natural waters, (2) collection in tanks when source
waters need to be transported over some distance and subsequent pumping from the tanks into the mesocosm, (3)
lowering a flexible bag like a curtain over an undisturbed water column. In all cases care should be taken to fill
the mesocosms with identical source waters. Considering that water masses may change over the filling procedure,
this can best be achieved by filling the mesocosms in parallel through a distributor system (Fig. 4). Likewise, if
several tanks are needed to obtain the required source water volume, the water of each tank should be distributed
evenly into all mesocosm units. The source water should be representative for the targeted ecosystem. This
concerns the depth at which the source water is collected and, when diurnally vertically migrating organisms are
present, the time of day. When pumping is applied some damage to fragile organisms, e.g. gelatinous zooplankton,
is unavoidable. It is therefore recommended to use pumps that ensure a smooth flow of pumped water, e.g.
peristaltic pumps (Fig. 4). To prevent large and rare organisms from entering and being unevenly distributed in
the mesocosms, some screening can be applied at the intake of the pumping hose.
As mentioned above a typical artifact of mesocosm enclosures is the reduced level or absence of turbulence. In
mesocosms with solid wall structures it may be useful to apply some form of mixing of the water column,
considering that turbulence (including its absence) is known to strongly affect the plankton community
composition and succession. In floating enclosures with flexible walls some turbulence is induced by surface wave
action, below surface water movement and variability in water currents, but the vorticity of the enclosed water is
still always much reduced compared to that of the natural environment. Somewhat related to the mixing regime
is another potential artifact in mesocosms where settling particulate matter is continuously resuspended from the
bottom. Resuspension of degrading organic matter, which under natural conditions would sink out of the upper
mixed layer, exaggerates the heterotrophic processes in the system. Collecting and removing the sedimented
matter in cone-shaped sediment traps which form the bottom of the mesocosms can avoid this problem (Fig. 4).
Benthic mesocosms
A particular challenge in benthic mesocosm experiments concerns the filling with sediment from the seafloor.
Depending on the size of the tanks and the sediment height, it may be necessary to transfer several hundreds of
kilograms of sediment from the seafloor to the tanks. Near intact sediments (undisturbed vertical stratification)
may be collected relatively easily in sub-tidal areas. At sea, undisturbed sediments may be retrieved using a box
corer or similar device, although this may be a tedious exercise involving multiple deployments of the coring
equipment. Large amounts of sediment can be gathered relatively easily and quickly using a sediment grab, but
disturbance of the sediment matrix is inevitable, and longer equilibration times for the sediment geochemistry to
stabilize will be required before experiments can be started. In any case, benthic communities within mesocosms
may be altered from those in natural ecosystems and a sound understanding of the equilibration period is crucial
to allow for changes in benthic communities and the establishment of a new steady state within the benthic
mesocosm. This equilibration period should be determined based on the specific conditions of the mesocosm
experiment, including the number of replicates, environmental parameters, and the selected organisms. Adequate
monitoring and sampling during the equilibration period are essential to ensure that the experimental conditions
have stabilized and the ecosystem has reached a new steady state which in turn increases material and labour
requirements. Robust control units are crucial in benthic mesocosm experiments and should ideally consist of the
same number of replicates as the treatment group to ensure that any observed changes are due to the experimental
treatments rather than natural variability. Sampling and monitoring should be in the same manner as the treatment
group.
**5 Alkalinity manipulation and monitoring**
Different minerals, waste materials and electrochemical products have been suggested as feedstock for ocean
alkalinity enhancement (for a comprehensive introduction to potential source materials see Eisaman et al. 2023).
Most source materials do not come as pure alkalinity, but contain other substances, such as silicate, calcium,
magnesium and various trace metals (e.g. iron, nickel, cobalt, chromium). OAE can be achieved by addition in
dissolved form, which requires dissolution of the feedstock before its release into the sea, or in particulate form,
after grinding of the feedstock, with the grain size being one important factor determining the dissolution rate.
OAE can further be conducted in a $CO_2$-equilibrated mode, which involves some form of active injection of $CO_2$
into the alkalinity-enriched source water prior to its release, or in a non-equilibrated mode, which relies on air-sea
gas exchange to provide the additional $CO_2$ that the alkalinized seawater can absorb. In case of the latter it is
important to keep in mind that the time scales for $CO_2$ equilibration are on the order of months and can only occur
as long as the alkalinized seawater is in contact with the atmosphere. (see Schulz et al., 2023 for further details)
Taken together, this results in a wide range of possible application scenarios, not all of which can be tested with
the same scrutiny in mesocosm experiments due to the high financial and personnel costs involved. Hence, it is
important to focus on those OAE application scenarios which are most likely to be implemented. As the field of
OAE R&D is developing rapidly and dynamically, there will likely be changes in what is considered the most
suitable OAE application approaches, in terms of cost, efficiency, environmental safety, friendliness in terms of
monitoring, verification and reporting (MRV), technological readiness, as well as the regulatory requirements for
their implementation. Mesocosm research in this field should maintain sufficient flexibility to respond to those
changes and aim for testing 'real-world' scenarios of OAE applications. On the other hand, because the results
obtained from mesocosm studies will likely be context-specific (depending on e.g. ecosystem type, time of year,
latitudinal location, hydrographic setting and depend on the mesocosm set-up and operation itself, it takes multiple
such studies for a given OAE approach to reach robust conclusions about its environmental safety. To facilitate
inter-comparison between results it would be favorable to use standardized mesocosms and follow common
protocols for mesocosm experimentation.
From an experimental perspective, there is a trade-off between testing pure alkalinity enhancement and feedstocks
which involve the release of other biologically active components. While the latter is more in line with real-world
applications, it complicates the interpretation of the observed responses due to confounding factors and limits the
extrapolation of the findings, considering that the stoichiometric composition differs between feedstocks. As the
field is currently still at an early stage and considering that the number of mesocosm studies will likely be small
due to their high costs, it seems beneficial to first establish a basic understanding of alkalinity effects in isolation,
before turning to more feedstock-specific testing. This being said, we note that the above-mentioned confounding
effects may actually be the intended research question or that the focus may be on a specific feedstock likely to
be utilized widely. In general, we recommend designing mesocosm experiments with a more generic approach
first and address feedstock-specific in smaller scale laboratory-based experiments.
Pelagic mesocosms
Alkalinity manipulations in pelagic mesocosms are fairly straightforward when done in dissolved form.
Dissolving the alkaline feedstock in freshwater or deionized water prevents secondary carbonate precipitation
during preparation of the concentrated solution (we note that the use of freshwater for feedstock dissolution may
not be practical for large-scale implementation of OAE). To avoid confounding effects of the freshwater addition
on the mesocosm community, the volume should be kept to a minimum. Using source materials with a high
solubility in water, such as $NaHCO_3$, $Na_2CO_3$, $Ca(OH)_2$ or NaOH enables highly concentrated alkaline source
water (Hartmann et al., 2023). To simulate $CO_2$-equilibrated alkalinisation $NaHCO_3$ and $Na_2CO_3$ can be combined
in appropriate proportions (Subhas et al., 2022), for non-equilibrated alkalinisation carbonate-free source
materials such as NaOH and $Ca(OH)_2$ can be used (Moras et al., 2021). To avoid prolonged pH peaks and
secondary precipitation during the injection procedure it needs to be assured that the concentrated solution is
mixed in rapidly. One way to achieve a uniform alkalinity enhancement across the water column is to move a
distribution device with multiple outlets up and down the mesocosms at a constant speed (Fig. 5). Flocculent
precipitates that form directly at the injection site are usually not stable and disappear quickly when further diluted
through mixing. Care should be taken to ensure that the added alkalinity is evenly distributed throughout the
enclosed water column.
Alkalinity enhancement in particulate form is far less practical. If the particles sink faster than they dissolve, they
accumulate on the mesocosm floor or sink directly into the trap in mesocosms with a sediment trap at the bottom.
Accumulation and subsequent dissolution at the bottom might lead to highly concentrated alkalinity enrichment,
enhancing the risk of secondary precipitation and of strong negative impacts in bottom waters. Alkaline particles
sinking into the sediment trap would be lost from the mesocosm enclosure during the next trap sampling. In both
cases it would be considered an experimental artifact. It is therefore recommended to use minerals with high
dissolution rates (e.g. NaOH, CaO, Ca(OH)$_2$, $_{Mg(OH)2}$) and small grain sizes to ensure dissolution before the mineral
particles reach the bottom of the mesocosms (see Eisaman et al. 2023 for a detailed description of technical aspects
of OAE).
Monitoring of seawater carbonate chemistry in the water column should adhere to the guidelines provided in
Schulz et al., 2023. High levels of non-equilibrated alkalinisation can lead to secondary precipitation, triggering
a process termed "runaway precipitation" (Moras et al., 2022; Hartmann et al., 2023), whereby carbonate
formation can consume more alkalinity than initially added. It seems that the initiation of this process can occur
both in the water column and on the mesocosm walls. As the carbonate crystals grow in size, their sinking velocity
increases. When incorporated in organic matter aggregates they serve as ballast, thereby increasing the vertical
flux of organic matter. In addition, carbonate crystals could affect mobility and feeding of plankton organisms,
with possible adverse effects on food web interactions and trophic transfer. Secondary precipitation also increases
seawater turbidity, affecting light attenuation and possibly primary production. Collecting this sinking particulate
matter in sediment traps at the bottom of the mesocosms enables the quantification and identification of the
precipitates and provides information about the chemical reactions leading to their formation. In mesocosms
without integrated sediment traps, simple traps can easily be set up on the bottom and sampled through a tube that
reaches the surface.
Benthic mesocosms
Alkalinity enhancement in the benthic mesocosm approach is achieved by mineral addition, which dissolves in
the surface sediment over time. In general, the addition of sedimentary OAE source materials (e.g. siliciclastic
minerals, carbonates; Eisaman et al., 2023) modifies the grain size distribution of the sediment and thus affects
the porosity, permeability, and water flow through the sediment. The changing sediment structure can impact
living conditions for organisms, as well as the distribution and abundance of organisms living in the sediment and
their behavior and ecology. With respect to mineral addition, the grain size selection is important, as a trade-off
between grain size and production costs is required (e.g. Hartmann et al., 2013). Previous studies have investigated
the relationship between $CO_2$-sequestration efficiency and grain sizes and there is a general assumption that small
grain sizes reveal higher dissolution rates and $CO_2$ sequestration rates due to larger reactive surface areas, whereas
more grinding energy is required generating a higher $CO_2$ footprint and lower $CO_2$-sequestration efficiencies
(Köhler et al., 2010; Renforth and Henderson, 2017; Foteinis et al., 2023). Clearly, the $CO_2$ emissions during
production and transport must be significantly lower than the potential $CO_2$ sequestration of benthic mineral
dissolution (see Eisaman et al., 2023). The selection of appropriate grain sizes for the addition of alkaline minerals
is a critical consideration for experimental studies, particularly in the context of the target environment's
geological setting. From an environmental perspective, it is recommended to choose comparable grain sizes that
are stable under in-situ hydrodynamic conditions. For highly dynamic ecosystems such as the Wadden Sea,
estuaries and wave-dominated coastal areas, a range of grain sizes from fine to coarse sand (0.075 to 2 mm) may
be appropriate for experimental approaches. However, in low-dynamic systems such as lagoons, enclosed bays,
or shelf regions, grain sizes from silt to very fine sand (<0.075 mm) can be considered for investigation. This
approach would also help to ensure that the sedimentary structure and settings for organisms in the mesocosms
are representative of the natural conditions of the target environment.
It may be practical to interrupt the water circulation system during mineral deployment in order to allow
sedimentation of the suspended matter. To achieve a uniform alkalinity enhancement in the benthic mesocosms,
minerals should be evenly distributed. To induce a measurable effect on alkalinity changes in the envisioned
experimental time, grain sizes smaller than 1 mm are desirable (Strefler et al., 2018). The addition to the marine
environment could best be achieved through a mixture of natural seawater, marine sediments, and OAE source
materials. This may ensure a more uniform distribution and reduce the purity of industrially produced OAE source
materials, which are poor in nutrients and microbial organisms. Thus, this approach is also recommended for the
addition of silicates to benthic mesocosms. By using a mixture, the potential effects of silicate addition can be
more accurately evaluated because the experimental conditions are more similar to those in the natural
environment.
For calcium carbonate, it may be reasonable to use the annual flux of POC to the seafloor as an upper estimate of
the required mineral to be added. The underlying assumption here is that the added mineral can completely
neutralize the natural $CO_2$ produced from organic matter degradation. However, this assumes that mineral
dissolution efficiency is close to 100 %, which may not be the case if it is mixed below the undersaturated layers.
Adding minerals in large excess risks clogging the surface layer and creating a physical barrier against effective
benthic-pelagic coupling of solute fluxes. Finding the optimal mineral dosage to achieve a balance between
dissolution efficiency and dissolution rate would likely be specific to the local environmental characteristics and
require testing at each potential mineral addition site. For silicate minerals (e.g. olivine), the upper limit of mineral
addition per square meter will also depend on the trace metal concentrations (Flipkens et al., 2021). Based on the
variation in Ni content of marine sediments (prior to the addition of olivine), this implies that the allowable range
for the addition of olivine is between 0.059 and 1.4 kg per square meter of seafloor without posing a risk to benthic
biota. This threshold is based on Environmental Quality Standards (EQS), which are derived from metal toxicity
data using methods such as species sensitivity distributions (SSDs). They provide threshold metal concentrations
in seawater or sediment that are considered protective for the aquatic environment and are used by industries,
governments, and environmental agencies to guide regulations. So far, these guidelines are only appropriate to
specific regions and environments and may need to be re-evaluated for a broader use in OAE applications.
Monitoring of mineral dissolution will be determined by the experimental design. A major drawback of a high
through-flow is that rapid dilution and flushing of geochemical tracers emitted from the sediment compromises
the analytical detection of dissolving alkaline minerals in the overlying water and the reliable assessment of the
effectiveness of the method (see also section 4.4.3). In this case, alternative ways of mineral dissolution detection
may be required. For instance, alkalinity enhancement may be detectable in pore fluids, which can be extracted
using filters (e.g., rhizones) inserted horizontally through holes pre-drilled vertically in the tank (Fig. 6). However,
the vertical sampling resolution may be too coarse to detect mineral dissolution close to the sediment surface.
Microelectrodes for $O_2$, pH and $H_2S$ are arguably a better alternative to detect changes in surface geochemistry in
the uppermost centimeters after mineral addition. An advantage of the high dilution factors is the potential
suppression of secondary mineral formation such as phyllosilicates and/or carbonates, that could reduce the net
$CO_2$-sequestration efficiency of OAE (Fuhr et al., 2022, Moras et al., 2022, Hartmann et al., 2023). Secondary
mineral formation is a common process in marine seafloor sediments, potentially impacting global carbon and
element cycles on a global scale, the controlling factors are not unambiguously identified to date (e.g. Rahman et
al., 2017; Torres et al., 2020; Geilert et al., 2023).
The deployment of benthic incubation chambers within the mesocosms themselves is a non-invasive method for
detecting alkalinity release following mineral addition (Fig. 6). These benthic chambers enclose a certain area of
the surface sediment and allow the accumulation of alkalinity and other components of interest over time, from
which benthic fluxes can be determined. Mineral dissolution rates can be estimated by comparison with control
mesocosms where no minerals were artificially added. Fluid sampling can be achieved by hand via suction using
connected tubing and syringes. Care is needed to prevent hypoxia or anoxia inside the chambers due to respiration
by benthic biota, which may be observable by a blackening of the sediment surface due to precipitation of iron
sulfide minerals. Low oxygen levels will result in an interruption to the normal respiration rates of animals causing
them to resurface. This may alter natural sediment mixing rates as well as mineral saturation states via changes in
biogeochemical turnover rates and pathways in the sediment. Together, these undesired artifacts may be reflected
in unrealistic fluxes of alkalinity and other solutes from the sediment. Completely interrupting the water flow to
the whole benthic mesocosm in order to detect changes in bottom water alkalinity will only serve to magnify these
side effects.
**Recommendations**
General
- Use inert materials for mesocosm hardware (e.g. plastics, stainless steel)
- Select the mesocosm size and experimental duration according to the enclosed community and processes
studied
- Choose the experimental design to maximize the statistical power and report it
- Maximize similarity in starting conditions between mesocosms during enclosure filling
- Monitor starting conditions before applying experimental treatment
- Allow for the natural (e.g. seasonal) succession and avoid out-of-season events
- Avoid confounding factors and perturbations other than the intended treatments
- Adapt the sampling frequency to the dynamics of the processes studied
- Determine spatial heterogeneity and take account of it in the sampling strategy
- Apply depth-integrated sampling in case of vertical gradients (pelagic mesocosms)
- Minimize wall growth, e.g. by regularly cleaning the walls
OAE-specific
- Test real-world OAE scenarios, focusing on those most likely to be implemented
- Keep some flexibility to respond to changes in the OAE R&D field
- Monitor carbonate chemistry with at least two carbonate system parameters and watch out for
secondary precipitation
- Maximize transferability of results by testing generic OAE approaches
- Take note of the context-specificity of the observed ecosystem responses
- Provide detailed information of the feedstock composition utilized for experimental manipulations
-   Closely monitor signs of potential barriers to OAE implementation, e.g. long-term restructuring of
497           community composition and functioning, decline in ecosystem productivity, proliferation of harmful
498           species, disruption of trophic transfer, changes in elemental cycling

**Competing interests**

None of the authors has any competing interests.

**Author Contributions**

UR scoped and edited the contents of the manuscript. UR drafted the general text, with contributions from all
co-authors. UR drafted the sections specific to pelagic mesocosms, with contributions from DB. SG, AD, and
MK drafted the sections specific to benthic mesocosms. All authors contributed to revising the manuscript.

**Acknowledgements**

This is a contribution to the "Guide for Best Practices on Ocean Alkalinity Enhancement Research". We thank
our funders the ClimateWorks Foundation and the Prince Albert II of Monaco Foundation. Thanks are also due
to the Villefranche Oceanographic Laboratory for supporting the lead authors' meeting in January 2023. UR
acknowledges funding from the European Union's Horizon 2020 Research and Innovation Program under grant
869357 (project OceanNETs: Ocean-based Negative Emission Technologies analysing the feasibility, risks, and
co-benefits of ocean-based negative emission technologies for stabilizing the climate). DB acknowledges funding
from the Prince Albert II of Monaco Foundation for the OACIS project "Ocean alkalinity enhancement: a
mesocosm-scale approach". SG and UR acknowledge funding from the German Federal Ministry of Education
and Research (Grant No 03F0895) Project RETAKE, DAM Mission "Marine carbon sinks in decarbonization
pathways" (CDRmare).

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

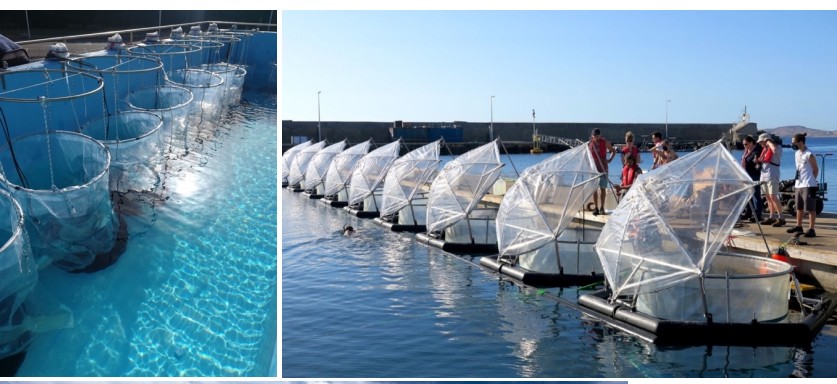

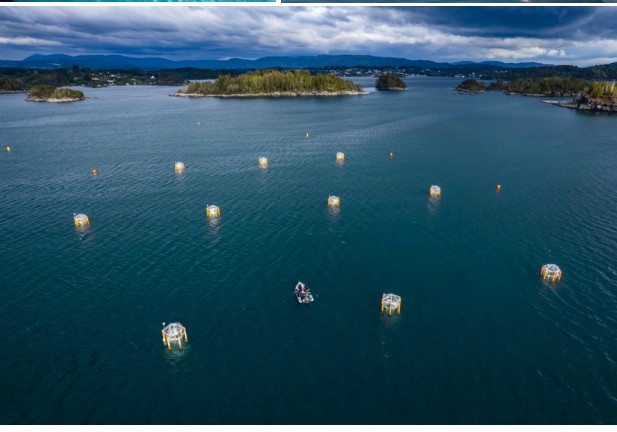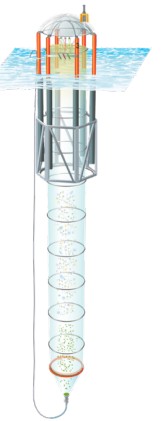

Figure 1. Pelagic mesocosm facilities currently used in OAE research. *top left:* Land-based mesocosms (1 m3) at the University of Vigo, Spain. *top right:* In situ on-shore mesocosms (10 m3) operated by GEOMAR, here employed on Gran Canaria, Spain. *bottom left:* Kiel Off-Shore Mesocosms for Ocean Simulations (KOSMOS), here employed in the Raunefjord, Norway. *bottom right:* Sketch of a KOSMOS mesocosm unit (55 m3). Photo/graphic sources: *ul:* Daniela Basso, University of Milano-Bicocca, *ur:* Ulf Riebesell, GEOMAR, *bl:* Uli Kunz, *br:* Rita Erven, GEOMAR.


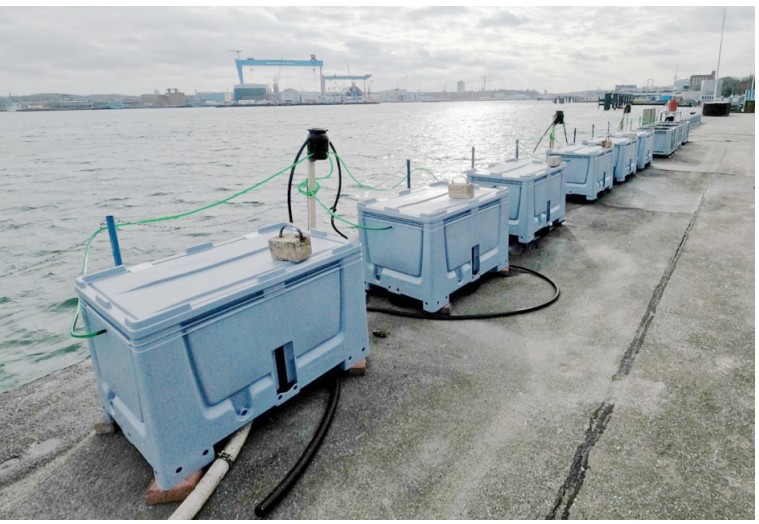


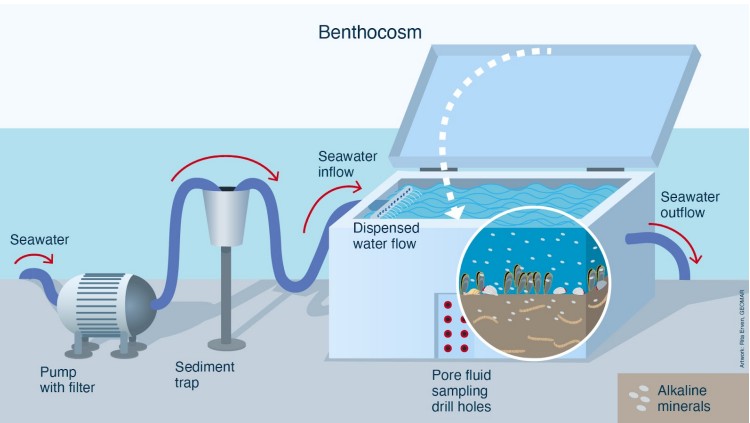

Figure 2. *top:* Benthic mesocosm units currently (2022-2023) installed at the Kiel Fjord, Germany. *bottom:*
Sketch of the experimental set-up for the benthic mesocosms shown in top picture. Photo/graphic source:  top:
Sonja Geilert; bottom: Rita Erven, GEOMAR.



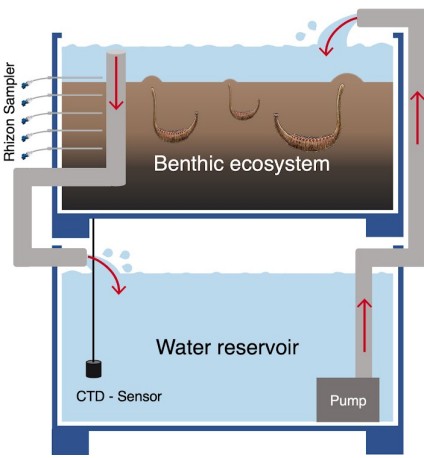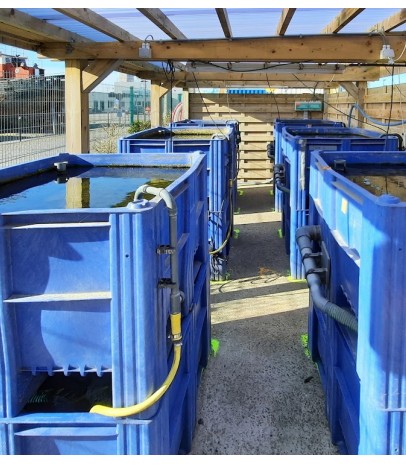

Figure 3: In the benthic mesocosms at the University of Antwerp the dissolution kinetics of silicate minerals and
the impacts on the benthic fauna in coastal environments are monitored since 2019. The system comprises 20
units with two stacked tanks, the upper tank is housing the benthic ecosystem, and the lower tank is functioning
as a water reservoir. Natural sediment of 40 sediment height with a mean grain size of 123 µm (3.0 phi) was
collected from an intertidal sand flat in the Oosterschelde (Netherlands) and mixed with olivine sand of similar
grain size. Water from the Easter Scheldt Estuary (salinity 32-35) is used to conduct flux-sessions of 5 weeks
(weekly sampling). At the end of each session, the total volume of water in each unit (~500 L) is renewed
(Drawing: A. Hylén, Photo: M. Kreuzburg https://www.coastal-carbon.eu/, Geobiology, University of Antwerp).







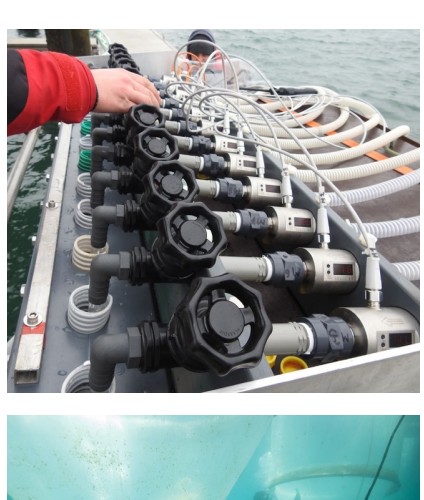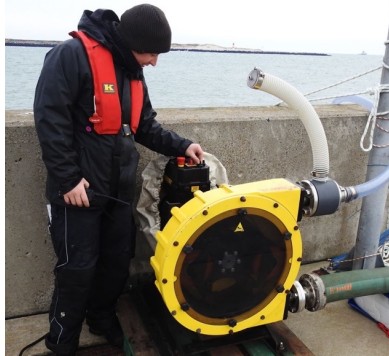

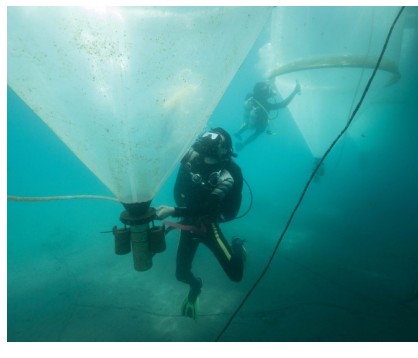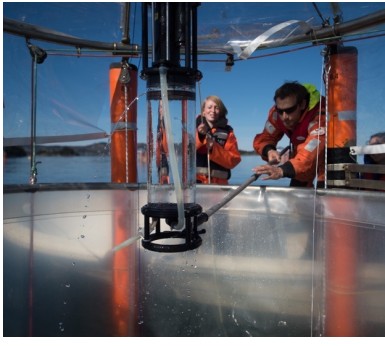


Figure 4: *Upper left:* Distributor control system enabling parallel filling of all mesocosms. *Upper right:* Peristaltic
pump ensuring smooth flow of source water during filling of the mesocosms, keeping damage to fragile organisms
at a minimum. *Lower left:* Sediment traps forming the bottom of in situ mesocosm enclosures. *Lower right:*
Programmable water sampler, enabling dept-integrated water samples over the entire mesocosm depth (or parts
thereof). (Photo sources: *ul, ur:* Ulf Riebesell, *ll:* Michael Sswat, *lr:* Solvin Zankl)



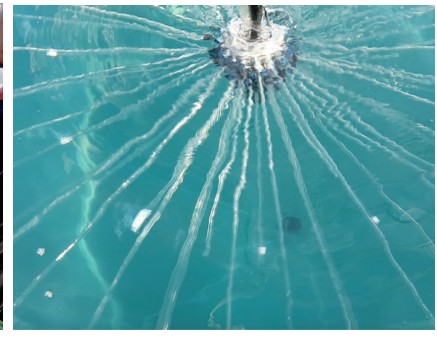

Figure 5: *Left:* Distribution device used for alkalinity addition; by moving it up and down in the water column
during alkalinity injection at constant speed a uniform alkalinity enhancement can be achieved. *Right:* Milky water
at the outlet of the injection tubes indicates temporary precipitation which, however, quickly disappears as the
highly concentrated alkalinity solution dilutes. Photo sources: Ulf Riebesell

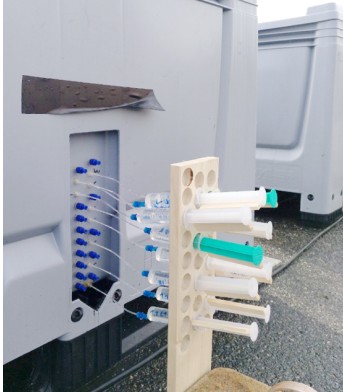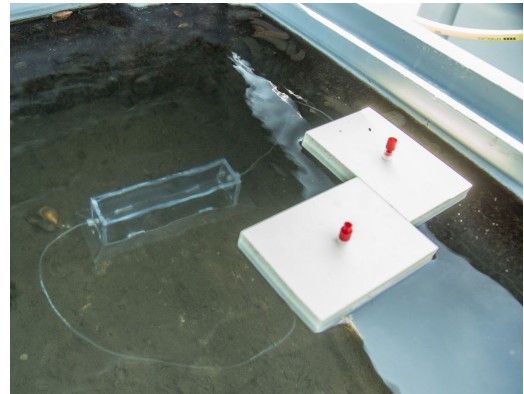

Figure 6. *left:* Pore fluid sampling using rhizons. *right:* benthic incubation chamber to assess alkalinity
enhancement with respect to mineral dissolution in benthic mesocosm experiments. Photo sources: left Sonja
Geilert, right Michael Fuhr, GEOMAR.