# Peer review of "Mesocosm experiments in ocean alkalinity enhancement"

_State of the Planet, 2023_

## Referee Comment (RC1)

**Mesocosm experiments in ocean alkalinity enhancement research**

This is a valuable manuscript to describe the general characteristics of the mesocosm approaches, as well as specific considerations to test the efficacy of OAE applications using pelagic and benthic mesocosms.

My major concern is about the possibility of study long-term (several years) effects of OAE applications in the marine systems using mesocosm approach. The authors are well aware about this limitation as they mentioned that "*Another drawback of mesocosm experiments is their limited duration, (Line 101-102) ...*". This aspect may not be related directly to this "mesocosm chapter" and probably it is discussed in other chapters, however to convince public opinion, it is necessary that no harm will be done to marine ecosystems at long-term too. Therefore, it could be good to be mentioned and discuss a little bit more about the need of studies on long-term effects of OAE on marine ecosystems in this chapter.

Otherwise, the manuscript is very fin. I've some comments, questions and suggestions which I present below that I hope will make the manuscript more precise.

**Line (L) 57-58:** Therefore, to cover the seasonal succession, it is needed to conduct a mesocosm experiment during several seasons. Could it be possible to maintain such mesocosm experiment during the year(s)? In any case, it could be good to mention something about this aspect as it was suggested above.

**L61:** inorganic nutrient concentration, I would suggest to add a "s" for concentration as there are several nutrients.

**L66:** *biological characteristics (e.g. ecosystem vulnerability, time of season).* System vulnerability depends on the communities present at the moment of the mesocosm experimentation. So, "communities" could be added as a first example of biological characteristics.

**L68-69:** *Key research questions which can be addressed adequately in mesocosm experiments are: …..* The first, second and last mentioned items are not the Key research questions which can be addressed in the mesocosm experiments, but they are requirements which can be established before mesocosm OAE experiment.

**L81:** I would suggest to use "communities" here in plural as there are several communities (bacterial, phytoplankton, zooplankton, etc.).

**L141-143:** addition of freshwater into the upper layer or brine into the bottom layer of a mesocosm, could be considered as a new treatment engendering marine organism responses to less or more saline water. This could provide additional complication for interpretation of the results regarding the effect of OAE.

**L161:** *1) What are alkaline mineral dissolution rates under ambient conditions?* As the results come from mesocosm experiment, "ambient conditions" could be replaced by "mesocosm conditions".

**L164-166:** Are there some published papers regarding these OAE benthic mesocosm experiments? If so, please mention them. The references were not mentioned for the OAE pelagic mesocosm, if there are some references for pelagic mesocosm, please mention them as well in the pelagic mesocosm's section.

**L169:** Please replace "filter" by "screen" and give an example of the size of screening.

**L175 :** Please give examples for the monitoring frequency: high frequency? every some hours? daily, weekly? etc.

**L177-182:** *Water circulation approach*. It could be useful to refer in the text, the number of the figures presented in the manuscript (Fig. 3). It was mentioned in L185, but it could be also indicated before.
As well, please also refer in the entire text to the number of the figures presented in the manuscript.

**L179:** *... a constant flow of water*. In the area with the tide water movement, the flow of water could/should be adjusted regarding the tide water movement? It is necessary or not?

**L186:** Please provide examples about replicates (at least 2?, 3?, > 3?).

**L192-193:** Please also give some insights about the type of sediments (sandy, rock, etc.) and benthic organisms that can be tested in the benthic mesocosms regarding OAE studies? All types of sediment and benthic organisms can be used and tested in the benthic mesocosms? If the experiment covers for example one year, how could be studied the "recruitment" of some benthic organisms that a part of their life happens in the pelagic system? It will be good if these aspects (types of sediment and benthic species) could be mentioned in the manuscript with some examples.

**L200-206:** *The time for equilibration may differ for pelagic and benthic habitats. Adequate monitoring during this pre-manipulation phase can determine when a new steady state is reached and confirm whether all mesocosms have similar starting conditions.*
This means that the T0 samples of all mesocosms should be taken and analyzed and if the results are similar for all mesocosms, thereafter the real manipulation can be started and monitored. Please provide the type of samples (physical, chemical and/or biological), with some examples, which can be taken and analyzed during this pre-manipulation phase. This information will also help better understanding of the L255-257.

**L212-213:** *Some species may even perform diurnal vertical migration, which also should be accounted for in the sampling strategy.* Which sampling strategy should be considered related to diurnal vertical migration of organisms? Sampling during the night? At which depths? Because "taking depth-integrated water samples » cannot help to study diurnal vertical migration.

**L214-221:** What are the conclusions of this paragraph? In the OAE mesocosm experiment, cleaning of the mesocosm walls can/should be done or not? If yes, this additional precipitation nuclei in the water column could not provide the artifacts in the experiment and result interpretation?

**L262:** Pleased named here these different minerals, waste materials and electrochemical products and provide the references.

**L277:** Please use entire words for MRV at the first use in the text (Monitoring, Reporting and Verification?).

**L285-293:** This paragraph is very important and recommendations are very reasonable and logic.

**L296-307:** Some references are welcome for this section.

**L314-315:** *It is therefore recommended to use minerals with high dissolution rates (e.g. CaO, Ca(OH)₂) and small grain sizes to ensure dissolution before the mineral particles reach the bottom of the mesocosms.* As the mesocosms are not very deep, and regarding to estimation of settling rate of these grains, what size these grains should have to dissolute before reaching the bottom of the mesocosm? It could be useful to provide in the text a notion about the grain sizes (less than ??) which are recommended to be use regarding the mesocosm deep.

**L318:** What could be the interaction of "secondary precipitation" with organisms in the water column of the mesocosm or in the mesocosm wall? It could be useful to provide some insights about this potential interaction(s), or if the effect of secondary precipitation on organisms is not clear, it can be mentioned in the text.

**L328:** Please identify the mineral which was added.

**L365:** *... addition of olivine is between 0.059 and 1.4 kg per square meter of seafloor without posing a risk to benthic biota.* How about the risk for the planktonic organisms? Are these values independent of the water column deep over the benthic biota? Are there the same for example if there are some cm or 1 m of water column over the sediment?

**L386-387:** *Care is needed to prevent hypoxia inside the chambers.* How does know the hypoxia occurred inside the chambers. By measuring continuously at high frequency oxygen concentrations inside the chambers? Otherwise?

**L501-504:** Please replace m3 by m³ in legend of Figure 1, and also show the scale for the bottom right figure (or mention the depth of the KSOMOS mesocosm unit in the legend).

**End of the review.**

Behzad Mostajir

July 11 2023

---

## Author Comment (AC1)

**Reviewer #1**

This is a valuable manuscript to describe the general characteristics of the mesocosm approaches, as well as specific considerations to test the efficacy of OAE applications using pelagic and benthic mesocosms.

My major concern is about the possibility of study long-term (several years) effects of OAE applications in the marine systems using mesocosm approach. The authors are well aware about this limitation as they mentioned that "*Another drawback of mesocosm experiments is their limited duration, (Line 101-102) …*". This aspect may not be related directly to this "mesocosm chapter" and probably it is discussed in other chapters, however to convince public opinion, it is necessary that no harm will be done to marine ecosystems at long-term too. Therefore, it could be good to be mentioned and discuss a little bit more about the need of studies on long-term effects of OAE on marine ecosystems in this chapter.

Otherwise, the manuscript is very fin. I've some comments, questions and suggestions which I present below that I hope will make the manuscript more precise.

*Response: On behalf of my co-authors I thank Behzad for this insightful review. Below a point by point response to his specific comments. Regarding his more general introductory note, we fully agree with Behzad's concern about the need to test for long-term (several years) impacts of OAE applications. This pertains to (a) small alkalinity changes at large spatial scales (e.g. at the basin-scale), (b) moderate alkalinity changes at regional scales (e.g. in marginal seas), and (c) potentially large alkalinity changes at local scales (e.g. at the point source of alkalinity release for geographically fixed release sites). For none of these valid research questions, mesocosm experiments provide a suitable tool, because of their limitations regarding experimental duration. The duration of such studies should ideally encompass time scales relevant for adaptive processes, such as evolutionary adaptation and community reorganization. Those studies, in the absence of long-term field experiments, are currently best done with natural analogs. In this respect, the paper by Subhas et al. on natural analogs to ocean alkalinity enhancement provides a good discussion on relevant time scales for OAE impact testing. In view of the well-known limited duration of mesocosm experiments we chose to refrain from including a discussion on long-term impacts.*

**Specific comments:**

Line (L) 57-58: Therefore, to cover the seasonal succession, it is needed to conduct a mesocosm experiment during several seasons. Could it be possible to maintain such mesocosm experiment during the year(s)? In any case, it could be good to mention something about this aspect as it was suggested above.

*Response: We agree and will address this by expanding the corresponding section.*

L61: inorganic nutrient concentration, I would suggest to add a "s" for concentration as there are several nutrients.

*Response: Will be done as requested.*

L66: *biological characteristics (e.g. ecosystem vulnerability, time of season)*. System vulnerability depends on the communities present at the moment of the mesocosm experimentation. So, "communities" could be added as a first example of biological characteristics.

*We will clarify this in our revision.*

**L68-69:** *Key research questions which can be addressed adequately in mesocosm experiments are: …..* The first, second and last mentioned items are not the Key research questions which can be addressed in the mesocosm experiments, but they are requirements which can be established before mesocosm OAE experiment.

*Response: We disagree with the reviewer on this point. Mesocosm experiments can address the question ´What is the safe operating space for OAE applications with respect to possible impacts on marine ecosystems functioning, biodiversity, and ecosystem services?´ and ´How could OAE be implemented to reduce the risk of inadvertent negative environmental effects, and maximize co-benefits?´ Those question have in fact been addressed in OAE mesocosm experiments which we, the authors of this manuscript, have conducted over the past few years. If done at different locations with diverging ecosystem in a comparable manner mesocosm experiments can also address the last research question in our list ´Which application sites are most appropriate for which OAE approach?´ We would therefore choose to leave the list of possible research questions unchanged.*

**L81:** I would suggest to use "communities" here in plural as there are several communities (bacterial, phytoplankton, zooplankton, etc.).

*Response: Will be done as suggested.*

**L141-143:** addition of freshwater into the upper layer or brine into the bottom layer of a mesocosm, could be considered as a new treatment engendering marine organism responses to less or more saline water. This could provide additional complication for interpretation of the results regarding the effect of OAE.

*Response: Point well taken. This optional experimental manipulation will be further explained.*

**L161:** *1) What are alkaline mineral dissolution rates under ambient conditions?* As the results come from mesocosm experiment, "ambient conditions" could be replaced by "mesocosm conditions".

*Response: We will follow the reviewer recommendation.*

**L164-166:** Are there some published papers regarding these OAE benthic mesocosm experiments? If so, please mention them. The references were not mentioned for the OAE pelagic mesocosm, if there are some references for pelagic mesocosm, please mention them as well in the pelagic mesocosm's section.

*Response: Unfortunately, the work referred to in this section has been published yet. We will modify this part taking the reviewer's comment into account.*

**L169:** Please replace "filter" by "screen" and give an example of the size of screening.

*Response: Will be done as suggested.*

**L175 :** Please give examples for the monitoring frequency: high frequency? every some hours? daily, weekly? etc.

*Response: This will be clarified in our revision.*

**L177-182:** *Water circulation approach*. It could be useful to refer in the text, the number of the figures presented in the manuscript (Fig. 3). It was mentioned in L185, but it could be also indicated before. As well, please also refer in the entire text to the number of the figures presented in the manuscript.

*Response: Will be done as suggested.*

**L179:** *… a constant flow of water*. In the area with the tide water movement, the flow of water could/should be adjusted regarding the tide water movement? It is necessary or not?

*Tidal cycles might influence the general alkalinity cycles due to current-induced advective fluxes. However, this experiment is designed to investigate the interactions of OAE source material compared to non-OAE source material present and fluxes are induced by bio-irrigation.*

**L186:** Please provide examples about replicates (at least 2?, 3?, > 3?).

*Response: Replicate number should be ≥3. This will be incorporated in the revised manuscript.*

**L192-193:** Please also give some insights about the type of sediments (sandy, rock, etc.) and benthic organisms that can be tested in the benthic mesocosms regarding OAE studies? All types of sediment and benthic organisms can be used and tested in the benthic mesocosms? If the experiment covers for example one year, how could be studied the "recruitment" of some benthic organisms that a part of their life happens in the pelagic system? It will be good if these aspects (types of sediment and benthic species) could be mentioned in the manuscript with some examples.

*Response: Thank you for pointing this out. We will expand on this in the revised manuscript.*

**L200-206:** The time for equilibration may differ for pelagic and benthic habitats. Adequate monitoring during this pre-manipulation phase can determine when a new steady state is reached and confirm whether all mesocosms have similar starting conditions.

This means that the T0 samples of all mesocosms should be taken and analyzed and if the results are similar for all mesocosms, thereafter the real manipulation can be started and monitored. Please provide the type of samples (physical, chemical and/or biological), with some examples, which can be taken and analyzed during this pre-manipulation phase. This information will also help better understanding of the L255-257.

*Response: We will expand on this in the revised manuscript.*

**L212-213:** *Some species may even perform diurnal vertical migration, which also should be accounted for in the sampling strategy.* Which sampling strategy should be considered related to diurnal vertical migration of organisms? Sampling during the night? At which depths? Because "taking depth-integrated water samples » cannot help to study diurnal vertical migration.

*Response: The sampling strategy needs to be adapted to the behavior of the migrating organisms. We think it is beyond the scope of this paper to specify sampling strategies for the various vertical migrators.*

**L214-221:** What are the conclusions of this paragraph? In the OAE mesocosm experiment, cleaning of the mesocosm walls can/should be done or not? If yes, this additional precipitation nuclei in the water column could not provide the artifacts in the experiment and result interpretation?

*Response: This is indeed a difficult decision in the situation where secondary precipitation occurs on the mesocosm walls. We will expand on this question in our revision and provide some guidance.*

**L262:** Pleased named here these different minerals, waste materials and electrochemical products and provide the references.

*Response: The reader is referred to the excellent overview on feedstock materials by Eisaman et al. 2023.*

**L277:** Please use entire words for MRV at the first use in the text (Monitoring, Reporting and Verification?).

*Response: Will be done as suggested.*

**L285-293:** This paragraph is very important and recommendations are very reasonable and logic.

*Response: We thank the reviewer for this positive feedback.*

**L296-307:** Some references are welcome for this section.

*Response: Thank you for pointing this out. Will be added during the revision.*

**L314-315:** *It is therefore recommended to use minerals with high dissolution rates (e.g. CaO, Ca(OH)$_2$) and small grain sizes to ensure dissolution before the mineral particles reach the bottom of the mesocosms.* As the mesocosms are not very deep, and regarding to estimation of settling rate of these grains, what size these grains should have to dissolute before reaching the bottom of the mesocosm? It could be useful to provide in the text a notion about the grain sizes (less than ??) which are recommended to be use regarding the mesocosm deep.

*Response: We believe this is beyond the scope of the paper. Also here we would refer the reader to the overview on feedstock materials by Eisaman et al. 2023.*

**L318:** What could be the interaction of "secondary precipitation" with organisms in the water column of the mesocosm or in the mesocosm wall? It could be useful to provide some insights about this potential interaction(s), or if the effect of secondary precipitation on organisms is not clear, it can be mentioned in the text.

*Response: We will expand on this in our revision of the manuscript.*

**L328:** Please identify the mineral which was added.

*Response: The added minerals will be specified in the revised manuscript.*

**L365:** *… addition of olivine is between 0.059 and 1.4 kg per square meter of seafloor without posing a risk to benthic biota.* How about the risk for the planktonic organisms? Are these values independent of the water column deep over the benthic biota? Are there the same for example if there are some cm or 1 m of water column over the sediment?

*Response: We will clarify this in our revision.*

**L386-387:** *Care is needed to prevent hypoxia inside the chambers.* How does know the hypoxia occurred inside the chambers. By measuring continuously at high frequency oxygen concentrations inside the chambers? Otherwise?

*Response: We will clarify this in our revision.*

**L501-504:** Please replace m3 by $m^3$ in legend of Figure 1, and also show the scale for the bottom right figure (or mention the depth of the KSOMOS mesocosm unit in the legend).

*Response: Will be done as suggested.*

---

## Author Comment (AC2)

**Reviewer #2**

This is a well written and very valuable overview of mesocosm approaches to OAE studies, with important information and considerations that should (need) to be accounted for in future studies. The authors should be commended for providing a clear overview. There are a few technical issues to be addressed, only minor suggestions to change wording slightly to clarify meaning and to ensure that key points are made clear.

*Response: On behalf of my co-authors I thank Alex for this positive and constructive review. See our point by point responses below.*

Ln 16: 'realism' – This seems unnecessarily contentious; mesocosms are indeed far nearer 'reality' than microcosms and cultures due to their inclusion of a more representative portion of the ecosystem, but it is only a portion. This is clearly reflected in latter parts of the review so a more appropriate term or just leaving it at 'biological complexity' may fit better in the abstract (e.g., 'By *combining representative biological complexity with controllability and replication..').*

*Response: We agree with the reviewer and will delete "realism".*

Ln 42: rather than 'poorly realistic', 'unrealistic'?

*Response: We find the term ´unrealistic´ too strong for laboratory culture experiments and would prefer to stick with our wording.*

Ln 123: 'relevant information in the context of regulatory considerations' – another important strength (of the gradient approach) is its utility to model parameterization in terms of functional responses of organism physiology and ecosystem function.

*Response: We thank the reviewer for this suggestion and will incorporate it in the revised manuscript.*

Ln 204-205: 'the time for equilibrium may differ for pelagic and benthic habitats' – is variability in the equilibrium time for different ecosystems and seasons another consideration that needs to be made?

*Response: We agree and will expand on this in the revised manuscript.*

Ln 281: '…)' – seems unnecessary to extend the list here.

*Response: Agreed. Will be deleted.*

Ln 356-357: It is not clear whether the authors mean rain ratio (PIC:POC) or sedimentation rate in this line. Based on the rest of the paragraph it is likely the former but this line is unclear. Please clarify and rephrase.

*Response: Point well taken. Will be clarified.*

Ln 364-367: Are the thresholds given here, based on 'Environmental Quality Standards', international or will they vary depending on regional authority?

*Response: Will be clarified.*

Ln 396: Can the authors give some examples of the inert materials that should be considered?

*Response: Will be added in the revised manuscript.*

Lns 407-414: These recommendations are really important to the fledgling field of OAE research; should they be incorporated into the abstract to ensure they are taken up by the research community? The fourth (transferability) and sixth (feedstock) are essential for the community to ensure consistent and value-for-money OAE research that advances the field rather than causing confusion.

*Response: In our concluding sentence of the abstract we refer the reader to the recommendations at the end of the paper. We believe that too many of the recommendations are important to the OAE research community and decided against highlighting specific ones in the abstract.*

---

## Author Comment (AC3)

**Reviewer #3**

There is a scientific and ideological divide on the development on oCDR methods. Nevertheless, it is clear that such methods are being considered, and several international bodies are working on guidelines for testing oCDR methods. In this context, the current article concerns a timely topic. The article focusses on the use of mesocosms within one realm of oCDRs, i.e. OAEs.

Specifically, the article looks at the applicability of mesocosm experiments to different OAE-techniques, in pelagic as well as in benthic ecosystems. An important rationale is the use of mesocosms for testing of risks of OAEs. The article is logical and well-structured. It contains clear and concise advice on the planning and execution of mesocosm experiments within this framework.

As other reviewers already provided a suite of more detailed comments, which I generally find well motivated for consideration, I only have a minor comment.

This comment concerns the concept of statistical power, which is mentioned or alluded to on several occasions, e.g., in sections 2 and 3:

**88 – 89** "Given the diverse range of natural processes encountered in mesocosm experiments, external influences may be challenging to control, necessitating a robust monitoring strategy to achieve statistical power by either treatment replication or treatment gradients"

**103 – 104** "The increasing variability between mesocosms in this process makes it increasingly difficult to identify treatment effects with statistical significance."

**120 – 121** "In fact, strong within-treatment variability can easily mask subtle treatment effects."

Despite this general recognition of the importance statistical power, this topic is not part of the Recommendations at the end of the article. Since testing of risks is considered by the authors to be an important rationale for mesocosm experiments on OAEs, it would seem logical to include reporting of statistical power as a general recommendation (rows **395 – 405**). If statistical power is not reported, it is difficult to draw meaningful conclusions on risks (and especially on the lack of them). **I suggest, therefore, that the recommendations highlight the importance of reporting statistical power**. I think that this minor addition would serve the manuscript and the intentions of the authors well.

There are also some very minor linguistic issues, which I believe will be taken care of in the editorial process.

In conclusion, I recommend the manuscript is accepted after minor revision.

*Response: We thank the reviewer for the suggestion to "highlight the importance of reporting statistical power". We agree about the importance of making explicit recommendations in this regard and will incorporate this in the revised manuscript.*

---

## Author Response (AR2)

Response to editor's request for minor revision

Editor: "My only additional minor suggestion would be to add to the list of future looking recommendations, an idea of one or two possible outcomes from mesocosm experiments that may suggest that perhaps the methdology is not safe for deployment at greater scale. What are the sorts of outcomes that would be sufficiently alarming to suggest that the approach should not be pursued and vice versa?"

*Response: Following the editor's request we have added the following text to the list of recommendations:* "Closely monitor signs of potential barriers to OAE implementation, e.g. long-term restructuring of community composition and functioning, decline in ecosystem productivity, proliferation of harmful species, disruption of trophic transfer, changes in elemental cycling"